# Experiences of recently diagnosed urban COVID-19 outpatients: A survey on patient worries, provider-patient interactions, and neutralizing monoclonal antibody treatment

Lindsey E. Fish[1,2☉*], Samantha C. Roberts[3☉], Tellen D. Bennett[4,5], Nichole E. Carlson[3], Mika K. Hamer[6], Bethany M. Kwan[7], Seth Russell[4], Adane F. Wogu[3], Matthew K. Wynia[2], Adit A. Ginde[7]

1 Department of General Internal Medicine, Denver Health and Hospital, Denver, Colorado, United States of America, 2 Department of Medicine, Division of General Internal Medicine, University of Colorado School of Medicine, Aurora, Colorado, United States of America, 3 Department of Biostatistics and Informatics, Colorado School of Public Health, Aurora, Colorado, United States of America, 4 Department of Biomedical Informatics, University of Colorado School of Medicine, Aurora, Colorado, United States of America, 5 Department of Pediatrics (Critical Care), University of Colorado School of Medicine, Aurora, Colorado, United States of America, 6 Center for Bioethics and Humanities, University of Colorado School of Medicine, Aurora, Colorado, United States of America, 7 Department of Emergency Medicine, University of Colorado School of Medicine, Aurora, Colorado, United States of America

☉ These authors contributed equally to this work.
* lindsey.fish@dhha.org

## Abstract

### Background

COVID-19 patients have experienced worry, altered provider-patient interactions, and options to use novel treatments, initially with neutralizing monoclonal antibodies (mAbs). Limited research has been performed on these aspects of the COVID-19 outpatient experience.

### Objective

This study aimed to evaluate the experiences of outpatients recently diagnosed with COVID-19, who were eligible for use of mAbs, during the diagnosis and treatment process based on sociodemographic and clinical factors.

### Methods

This was a self-reported cohort study performed via telephone surveys. Participants included COVID-19 outpatients who met at least one emergency use criterion for mAbs during the first 120 days after a SARS-CoV-2 positive test. We analyzed survey results using multivariable logistic regression for non-scale outcomes and adjusted proportional odds logistic regression for scaled outcomes.

**Data availability statement:** Data may be available through the authors/institution if in compliance with HIPPA, institutional data access, and Colorado Institutional Review Board for researchers who meet the criteria for access to confidential data. We can collaborate with someone external via a process that entails a reasonable request and determination by the study steering committee (or in their absence, the Colorado Clinical and Translational Sciences Insitute at the University of Colorado). We cannot publicly share data due to data sharing policies and data use agreements in place with participating sites. Data requests should be sent to Dr. Adit A. Ginde, adit.ginde@cuanschutz.edu. If unavailable, please contact CCTSI at CCTSI. Helpdesk@ucdenver.edu, phone 720.848.7100.

**Funding:** Funded by the National Center for Advancing Translational Sciences of the National Institutes of Health [grant numbers UL1TR002535, UL1TR002535-03S3, UL1TR002535-04S2, and 1UM1TR004399-01A1] and supported by the Health Data Compass Data Warehouse project (healthdata-compass.org). The funders had no role in study design, data collection and analysis, decision to publish, or preparation of the manuscript.

**Competing interests:** I have read the journal's policy and the authors of this manuscript have the following competing interests: Dr. Ginde reports grants from the National Institutes of Health (NIH) for the conduct of this study and other grants related to COVID-19 from the NIH, Centers for Disease Control and Prevention, Department of Defense, AbbVie, and Faron Pharmaceuticals, outside the submitted work. The other authors have no competing interests to report. This does not alter our adherence to PLOS ONE policies on sharing data and materials.

## Results

Greater worry about their COVID-19 diagnosis was reported by younger, female, and Hispanic patients and those with Medicaid insurance, two or more comorbid conditions, BMI > 25, and at least 2 COVID-19 vaccinations. Greater provider trust was reported by patients with ≥ 2 years of college education, one or more comorbid conditions, and one or more COVID-19 vaccinations; whereas less provider trust was reported by patients ages 45–64 years, with usual place of care in a walk-in clinic, and those without Commercial, Medicare, or Medicaid insurance. In patients who did not receive mAbs, patients with Medicaid and those without Commercial/Medicare insurance were among the factors that were less likely be offered mAbs by a provider.

## Discussion

This report describes factors associated with multiple aspects of outpatients' experience of COVID-19. This study demonstrated that there are important differences in the experience of outpatient COVID-19 patients based on sociodemographic factors and clinical factors, as well as where additional strategies are needed to improve this experience and associated outcomes.

## Introduction

Patients with recently diagnosed COVID-19 have varied experiences of emotional worry, interactions with health care professionals, and considerations regarding novel treatments such as neutralizing monoclonal antibodies (mAbs) [1–40]. The COVID-19 pandemic has led to an increase in mental health problems including fear of infection, uncertainty, mood disorders including anxiety and depression, and sleep problems [1–3]; and fear of COVID-19 infection is known to vary by demographic and clinical factors [4–12]. High fear of COVID-19 illness has also been associated with decreased access to healthcare [13]. However, the extent to which patients worry about adverse outcomes once diagnosed with COVID-19 has not been measured, and it is not known whether worry or other factors affect patient behaviors including seeking medical consultation, seeking and/or accepting treatment, and conforming to other medical recommendations.

When seeking medical consultation, patients rely on their providers to direct their care. The provider-patient relationship is complex and based on multiple factors including verbal and non-verbal communication, information sharing between both parties, empathy, and decision-making [14]. Mutual trust within that relationship is critical to success and prior to the COVID-19 pandemic, trust in medical scientists was trending upwards [15]. The impact of the COVID-19 pandemic on the provider-patient relationship is unclear; however, most studies demonstrate an adverse impact on this relationship including decreased trust, with a few exceptions [16–20]. However, it is unknown if there is an impact on patients' trust in their providers once personally diagnosed with COVID-19.

mAbs were the first authorized antiviral therapeutics available for high-risk outpatients with COVID-19 to reduce the risk of severe illness, hospitalization, and death [21–31]. Unfortunately, uptake of mAbs was extremely low, with only 7.2% of Medicare beneficiaries receiving treatment [32]. In the broader population, well below 1% of eligible COVID-19 patients received mAbs [33], and even lower rates were observed for patients in rural areas, those of non-white race/ethnicity, non-English language speakers, and those with fewer social supports, low health literacy, high economic concerns, and/or limited access to a health care facility [34–38]. Among healthy community members, mAbs are generally viewed positively and most people indicate being open to receiving treatment [39]. Among those who received mAbs, most expressed satisfaction with their overall experience, communication with providers, and the quality-of-care provided [40]; however less is known of the experiences of mAb untreated patients.

This study aimed to explore the experiences of outpatients recently diagnosed with COVID-19, who were eligible for the use of mAbs, immediately after the diagnosis and treatment process including experiences of worry, interactions with providers, and decisions about treatment with mAb, based on sociodemographic and clinical factors.

## Methods

### Study design and participants

As part of a longitudinal prospective cohort study, we surveyed adults with a documented SARS-CoV-2 positive test or mAb administration within 120 days of their SARS-CoV-2 positive test date. This manuscript's specific survey questions and results were part of the initial survey administered. Patients were selected for survey administration using the electronic health record at two health systems in Colorado: UCHealth and Denver Health (S1 File). A third-party contract research organization contacted selected patients for potential study participation and obtained informed consent verbally via a script with verbal acceptance over the phone and documented this prior to survey administration. The Colorado Multiple Institutional Review Board (IRB# 21−2935) approved the study (including the consent process), and all study procedures were per the institution's ethical regulations according to the Declaration of Helsinki (1975). Contact for the survey was between April 20, 2021 and December 27, 2021. Participants could complete up to three consecutive surveys but at least one survey needed to be completed to be included in this study with a minimum of 14 days between surveys. Participants were eligible for a single survey if enrollment and survey administration occurred more than 77 days after their positive SARS-CoV-2 test date. They were eligible for two surveys when enrollment and first survey administration occurred > 35 days and <= 77 days from their SARS-CoV-2 test date, and eligible for three surveys when enrollment and first survey administration occurred <=35 days from their SARS-CoV-2 test date.

The survey included questions on participants' demographic information, COVID-19 treatment, COVID-19 symptoms, hospitalization, COVID-19 vaccination status, experience with mAbs, and COVID-19 disease and provider experience (S1 File). The survey was developed via expert consensus by the study investigators and was pilot tested and revised amongst study staff and then potential study participants to ensure clarity and face validity. This was done in an expedited process due to the pandemic induced public health emergency.

We excluded participants for the following reasons: 1) no reported COVID-19 symptoms after positive SARS-CoV-2 test date (N = 11); 2) mAb treatment status (treated or untreated) not reported (N = 14); 3) survey answers indicated the individual was not eligible to receive mAb treatment based on the established emergency use authorization (EUA) criteria (N = 207); or 4) were contacted more than 120 days after their positive SARS-CoV-2 test or mAb administration date (N = 10).

### Outcomes

For this study, we were interested in the participants' experience of their SARS-CoV-2 infection, including their level of worry when diagnosed with COVID-19, their provider-patient relationship during the diagnosis and treatment process, and

factors influencing whether they received mAbs. From the larger survey, we analyzed the responses to eight questions (S1 File); four had Likert scale responses with a scale from 1 (lowest) to 10 (highest), and four had categorical responses.

The Likert Scale responses were grouped (responses 1–3, 4–6, 7–10) to create a spread and make the responses statistically meaningful. For the following questions: "On a scale of 1 to 10, how worried were you when you learned you have COVID-19?" and "On a scale of 1 to 10, how worried were you about getting the mAb treatment?", we grouped responses to not worried (1–3), moderately worried (4–6), and extremely worried (7–10). For the question: "On a scale of 1 to 10, how difficult was it for you to get the mAb treatment?", we grouped responses to not difficult (1–3), moderately difficult (4–6), and extremely difficult (7–10). For the question, "If you had a close friend or relative who got COVID-19 and they were eligible to receive the mAb treatment…on a scale of 1 to 10, how likely would you be to recommend it?", we grouped responses to not likely (1–3), moderately likely(4–6), and extremely likely (7–10).

The categorical responses were dichotomized to make the responses statistically meaningful. For the question, "Do you think you were treated fairly when it comes to COVID-19-related care?", we dichotomized responses to treated fairly and not treated fairly. For questions regarding trust in doctors ("In general, how much do you trust that doctors or providers will do what is right when it comes to your COVID-19-related care?" and "How much do you trust that your doctor or provider will do what is right when it comes to your COVID-19-related care?") we dichotomized responses to trusting doctor and not trusting doctor. For the question, "To better understand why you did not receive the mAb treatment for COVID-19…do any of the following reasons apply to you?", we dichotomized responses to offered mAbs and not offered mAbs.

## Statistical analysis

We summarized the baseline sociodemographic characteristics of survey participants, mAb treatment status, and survey responses using descriptive statistics (mean and standard deviations (SD) for continuous variables, frequency and percentage for categorical variables) by treatment status (i.e., mAb treated vs. no mAb treatment groups) (S1 File). We assessed the association between our sociodemographic and clinical factors of interest and the categorical outcomes using multivariable logistic regression. We used multivariable proportional odds logistic regression to assess the association between our sociodemographic and clinical factors of interest and the Likert scale outcomes. To address potential confounding, we conducted adjusted analysis including all sociodemographic and clinical factors in our models. We also developed an analysis with baseline sociodemographic factors only (S1 File). Sociodemographic factors included age, gender, race/ethnicity, level of education, usual place to receive healthcare, and health insurance. Clinical factors included the number of comorbid conditions, BMI (using standard calculation in the EHR), and COVID-19 vaccination status. We used R Statistical Software [41] (version 3.6.0; R Foundation for Statistical Computing Vienna, Austria) to conduct analyses.

## Results

### Survey sample

The survey sample consisted of 1612 adult participants, 539 mAb treated and 1073 untreated patients. Those that received mAb treatment were older [mean = 59 (SD = 15) vs. 51 (SD = 16) years], more likely to be non-Hispanic White (81.6% vs. 71.2%), more immunocompromised (18.9% vs 10.7%), and had at least two COVID-19 vaccinations (41.6% vs. 24.4%) (Table 1).

### Worry about COVID-19 diagnosis

Evaluation of the participant's worry about their COVID-19 diagnosis was performed using a question regarding worry (S1 File). Greater worry about a COVID-19 diagnosis was reported in female and Hispanic patients and those with, Medicaid insurance, one or more co-morbid medical conditions, BMI > 25, and at least two COVID-19 vaccinations (Table 2, Fig 1). The largest effect was having two or more co-morbid conditions with an adjusted odds ratio (aOR) of 2.01 (confidence

**Table 1. Sociodemographic characteristics and clinical conditions of mAb treated and mAb untreated survey participants.**

|  | No mAb treatment (N = 1073) | mAb treated (N = 539) |
|---|---|---|
| Age in years, mean (SD) | 51 (16) | 59 (15) |
| 18-44 | 419 (39.0%) | 101 (18.8%) |
| 45-64 | 391 (36.4%) | 202 (37.5%) |
| ≥65 | 263 (24.5%) | 235 (43.7%) |
| Gender-Female | 542 (50.5%) | 286 (53.1%) |
| Race/Ethnicity |  |  |
| Non-Hispanic White | 764 (71.2%) | 440 (81.6%) |
| Non-Hispanic Black | 77 (7.2%) | 16 (3.0%) |
| Hispanic | 182 (17.0%) | 61 (11.3%) |
| Other | 20 (1.9%) | 7 (1.3%) |
| Level of Education |  |  |
| ≥2-year college | 663 (61.8%) | 374 (69.4%) |
| <2-year college | 386 (36.0%) | 154 (28.6%) |
| (Missing) | 24 (2.2%) | 11 (2.0%) |
| Usual Place of Care |  |  |
| Regular Doctor | 802 (74.7%) | 467 (86.6%) |
| Walk-In Care | 263 (24.5%) | 71 (13.2%) |
| Health Insurance |  |  |
| Commercial | 644 (60.0%) | 319 (59.2%) |
| Medicaid | 156 (14.5%) | 45 (8.3%) |
| Medicare | 185 (17.2%) | 162 (30.1%) |
| None/Other | 73 (6.8%) | 9 (1.7%) |
| Immunocompromised | 115 (10.7%) | 102 (18.9%) |
| Body Mass Index (BMI) |  |  |
| BMI < 25 | 204 (19.0%) | 117 (21.7%) |
| BMI >= 25 | 862 (80.3%) | 418 (77.6%) |
| Number of Other Comorbid Conditions |  |  |
| None | 514 (47.9%) | 172 (31.9%) |
| One | 324 (30.2%) | 187 (34.7%) |
| Two or more | 235 (21.9%) | 180 (33.4%) |
| COVID-19 Vaccination |  |  |
| None | 667 (62.2%) | 239 (44.3%) |
| One | 131 (12.2%) | 54 (10.0%) |
| At Least 2 | 262 (24.4%) | 224 (41.6%) |

Legend: Sociodemographic characteristics and clinical conditions of mAb treated and mAb untreated survey participants. Some variables have missingness, which are not reported as separate categories if they are less than 5%, mAb – monoclonal antibody.

interval [95%CI]: 1.51–2.68; p < 0.001) times that of those who did not have any co-morbid conditions (Table 2). The smallest significant effect was receiving at least two COVID-19 vaccinations with an aOR of 1.33 (95%CI: 1.04–1.71) times that of those who were unvaccinated (Table 2).

## Provider-patient relationship

Evaluation of the provider-patient relationship was performed using questions regarding fair treatment (S1 File) and trust of the provider (S1 File). Fair treatment was reported among patients who had at least two COVID-19 vaccinations

**Table 2. Adjusted odds ratio for worried about COVID-19 diagnosis.**

| | OR | 95% CI |
|---|---|---|
| **Age in Years** | | |
| 18-44 years | NA | |
| 45-64 years | 0.82 | (0.62, 1.08) |
| 65+years | 0.60 | (0.44, 0.82) |
| **Gender** | | |
| Male | NA | |
| Female | 1.92 | (1.54, 2.40) |
| **Race/Ethnicity** | | |
| Non-Hispanic White | NA | |
| Non-Hispanic Black | 1.35 | (0.82, 2.24) |
| Hispanic | 1.64 | (1.18, 2.29) |
| Other | 2.27 | (0.87, 6.19) |
| **Level of Education** | | |
| < 2 yr college | NA | |
| ≥ 2 yr college | 0.93 | (0.73, 1.18) |
| **Usual Place of Care** | | |
| Regular Doctor | NA | |
| Walk-In Care | 1.01 | (0.76, 1.35) |
| **Health Insurance** | | |
| Commercial/Medicare | NA | |
| Medicaid | 1.69 | (1.16, 2.48) |
| None/Other | 1.25 | (0.73, 2.13) |
| **Number of Comorbid Conditions** | | |
| None | NA | |
| One | 1.37 | (1.06, 1.78) |
| Two or more | 2.01 | (1.51, 2.68) |
| **Body Mass Index (BMI)** | | |
| BMI < 25 | NA | |
| BMI>=25 | 1.38 | (1.06, 1.81) |
| **Vaccination Status** | | |
| No Vaccine | NA | |
| One Vaccine | 0.92 | (0.63, 1.35) |
| At Least 2 Vaccines | 1.33 | (1.04, 1.71) |

Legend: Adjusted Odds Ratio for Worried about COVID-19 Diagnosis based on response to the question "On a scale of 1 to 10, how worried were you when you learned you have COVID-19?" with responses grouped into not worried (1–3), moderately worried (4–6), and extremely worried (7–10). OR, odds ratio; CI, confidence interval.

(aOR=1.88, 95%CI: 1.11–3.32) compared to those who were unvaccinated (Table 3). Trust in general for providers to do the right thing when it comes to patient COVID-19 care varied. Lower trust in their provider was reported among those ages 45−64 years old and those receiving primarily walk-in care; whereas higher trust in their provider for COVID-19 care was reported among patients with one or two co-morbid conditions and those who received one or at least 2 vaccinations. Lower trust in providers to do the right thing was reported in patients with a usual place of care as a walk-in clinic (aOR=0.62, 95%CI: 0.43–0.90) compared to those patients with a regular doctor (Table 3). Higher trust in providers to do

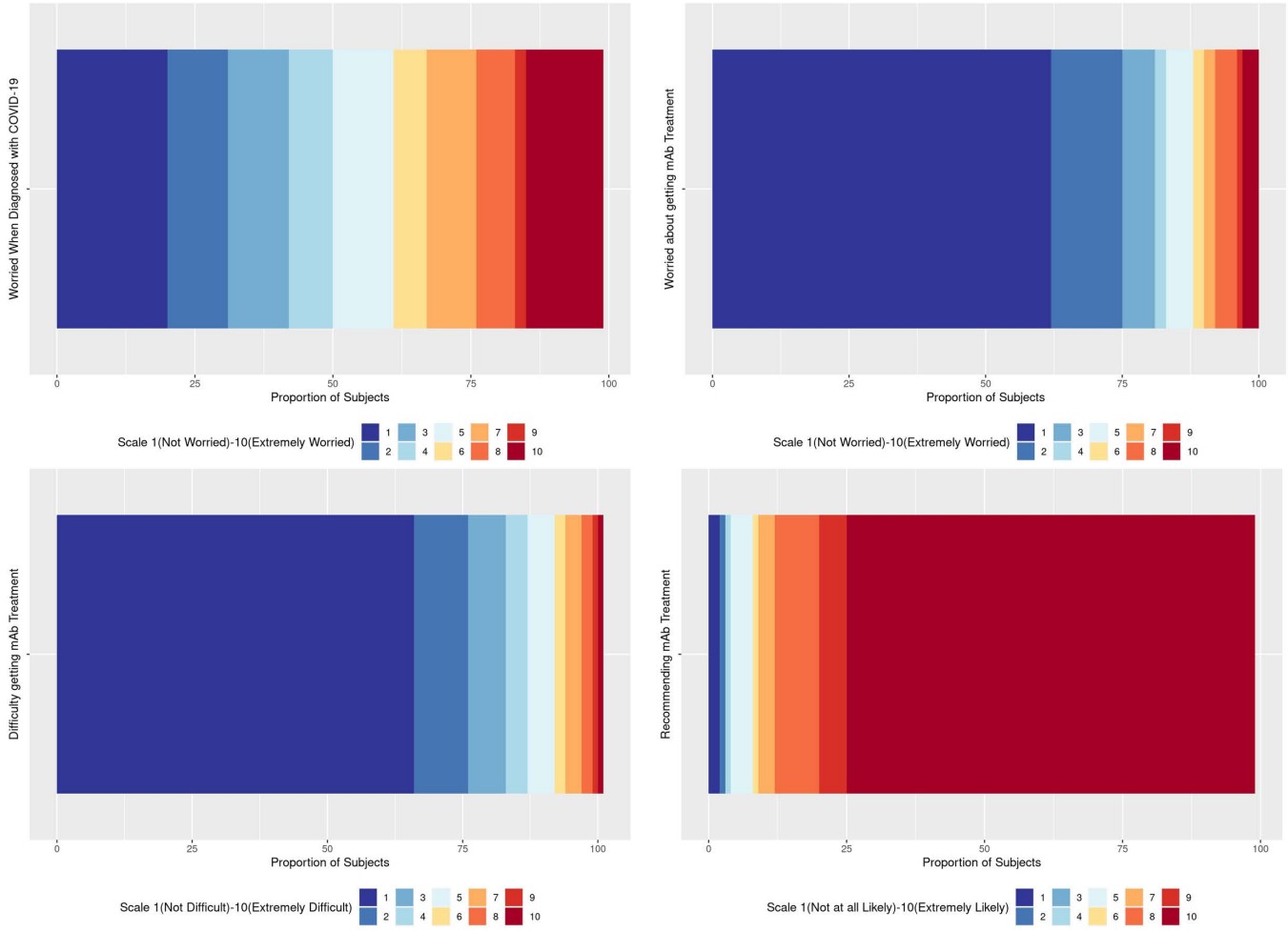

**Fig 1. Survey responses to worry about COVID-19 and worry, difficulty getting and recommending monoclonal antibody treatment.** Responses to questions "On a scale of 1 to 10, how worried were you when you learned you have COVID-19?", "On a scale of 1 to 10, how worried were you about getting the mAb (monoclonal antibody) treatment?", " On a scale of 1 to 10, how difficult was it for you to get the mAb treatment?", and "If you had a close friend or relative who got COVID-19 and they were eligible to receive the mAb treatment…on a scale of 1 to 10, how likely would you be to recommend it?".

the right thing regarding COVID-19 care was reported among patients with one comorbid condition compared to those that did not have any co-morbid conditions (aOR=1.59, 95%CI: 1.11–2.29). Higher trust in their provider was reported by patients who had a regular provider and greater than 2 years of college education (aOR=2.94, 95%CI: 1.35–6.41) compared to those with less than 2 years of college. Lower trust in their provider was reported in patients who had a regular provider but whose usual place of care was a walk-in clinic (aOR=0.34, 95%CI: 0.14–0.87) compared to those whose usual place of care was with their regular doctor (Table 3, Fig 2).

## Treatment with monoclonal antibody

Evaluation of the participant's experience with mAb was performed using questions regarding several aspects of treatment (S1 File). Lower odds of not being offered mAbs was reported among patients aged ≥65 and those with two or more COVID-19 vaccinations (aOR=0.56, 95%CI: 0.35–0.89; aOR=0.61, 95%CI: 0.42–0.89); whereas higher odds of

**Table 3. Adjusted odds ratio for trust in doctors for COVID-19 related care and treated fairly by doctors.**

| | Trust Doctor General | | Trust Main Doctor | | Treated Fairly | |
|---|---|---|---|---|---|---|
| | OR | 95% CI | OR | 95% CI | OR | 95% CI |
| **Age in Years** | | | | | | |
| 18-44 years | NA | | NA | | NA | |
| 45-64 years | 0.64 | (0.44, 0.93) | 1.34 | (0.58, 3.08) | 0.85 | (0.53, 1.33) |
| 65 + years | 0.85 | (0.54, 1.35) | 2.16 | (0.79, 6.22) | 1.48 | (0.82, 2.74) |
| **Gender** | | | | | | |
| Male | NA | | NA | | NA | |
| Female | 0.91 | (0.66, 1.24) | 0.78 | (0.37, 1.59) | 1.08 | (0.73, 1.60) |
| **Race/Ethnicity** | | | | | | |
| Non-Hispanic White | NA | | NA | | NA | |
| Non-Hispanic Black/Other | 1.33 | (0.74, 2.55) | 4.23 | (0.80, 78.54) | 1.02 | (0.52, 2.15) |
| Hispanic | 1.16 | (0.76, 1.80) | 4.24 | (1.32, 19.52) | 0.89 | (0.54, 1.52) |
| **Level of Education** | | | | | | |
| < 2 yr college | NA | | NA | | NA | |
| ≥ 2 yr college | 1.28 | (0.92, 1.78) | 2.94 | (1.35, 6.41) | 1.29 | (0.84, 1.96) |
| **Usual Place of Care** | | | | | | |
| Regular Doctor | NA | | NA | | NA | |
| Walk-In Care | 0.62 | (0.43, 0.90) | 0.34 | (0.14, 0.87) | 0.65 | (0.41, 1.03) |
| **Health Insurance** | | | | | | |
| Commercial/Medicare | NA | | NA | | NA | |
| Medicaid | 0.76 | (0.48, 1.22) | 0.78 | (0.29, 2.25) | 0.69 | (0.40, 1.21) |
| None/Other | 0.64 | (0.35, 1.23) | 1.89 | (0.33, 36.11) | 0.77 | (0.35, 1.88) |
| **Number of Comorbid Conditions** | | | | | | |
| None | NA | | NA | | NA | |
| One | 1.59 | (1.11, 2.29) | 1.15 | (0.52, 2.59) | 0.97 | (0.61, 1.55) |
| Two or more | 1.50 | (1.01, 2.26) | 1.70 | (0.68, 4.54) | 0.80 | (0.48, 1.32) |
| **Body Mass Index (BMI)** | | | | | | |
| BMI < 25 | NA | | NA | | NA | |
| BMI>=25 | 1.13 | (0.77, 1.66) | 0.44 | (0.12, 1.22) | 0.99 | (0.58, 1.64) |
| **Vaccination Status** | | | | | | |
| No Vaccine | NA | | NA | | NA | |
| One Vaccine | 2.23 | (1.32, 3.98) | 1.82 | (0.59, 8.00) | 1.63 | (0.90, 3.16) |
| At Least 2 Vaccines | 4.22 | (2.77, 6.62) | 1.05 | (0.44, 2.72) | 1.88 | (1.11, 3.32) |

Legend: Adjusted Odds Ratio for Trust in Doctor for COVID-19 Related Care and Treated Fairly by Doctors. Responses to questions "Do you think you were treated fairly when it comes to COVID-19-related care?", "In general, how much do you trust that doctors or providers will do what is right when it comes to your COVID-19-related care?", and "How much do you trust that your doctor will do what is right when it comes to your COVID-19-related care?". Responses were dichotomized into treated fairly and not treated fairly, and trusting doctor and not trusting doctor. OR, odds ratio; CI, confidence interval

not being offered mAbs was reported among patients with Medicaid insurance and none/other insurance (Table 4, Fig 1). None/Other insurance had the highest effect with a 3.01 (95%CI: 1.26–8.92) higher odds of not be offered mAbs and Medicaid insurance had the weakest with a 1.88 (95%CI: 1.06–3.50) higher odds of not being offered mAbs compared to those with Commercial/Medicare insurance. More difficulty to receive mAbs was reported in female patients (aOR=1.79, 95%CI: 1.08–3.00) compared to males (Table 4). Less difficulty to receive mAbs was reported among Medicaid patients (aOR=0.20, 95%CI: 0.03–0.73) compared to patients who had Commercial/Medicare insurance. Although not statistically

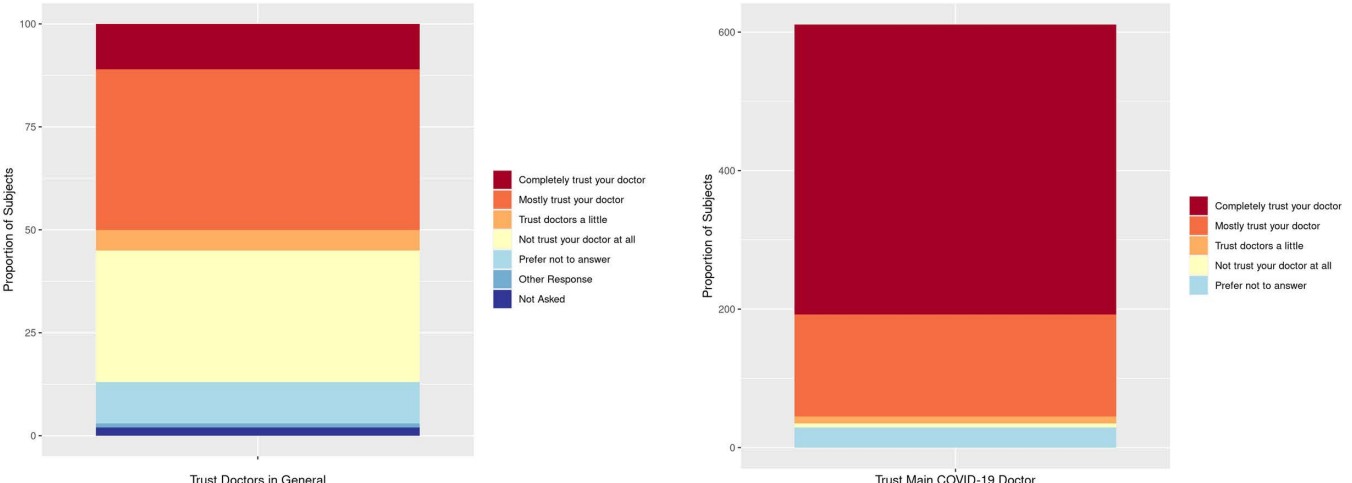

**Fig 2. Survey responses for trust in providers.** Responses to questions "In general, how much do you trust that doctors or providers will do what is right when it comes to your COVID-19-related care?" and "How much do you trust that your doctor will do what is right when it comes to your COVID-19-related care?".

significant, Hispanic patients were more likely to recommend mAbs to friends or relatives compared to Non-Hispanic White patients (aOR=2.73, 95%CI: 0.77–17.39 (Table 4, Fig 1)).

## Discussion

This report describes multiple aspects of outpatients' experience of COVID-19, including worry when diagnosed with COVID-19, provider-patient relationships during the diagnosis and treatment process, and treatment with monoclonal antibodies. This begins to fill a gap in our understanding of outpatient COVID-19 experiences. This study demonstrated that there are some important differences in the experience of COVID-19 based on sociodemographic and clinical factors; which indicates where additional strategies are needed to improve this experience, address disparities, and improve equity in receipt of antiviral treatment for COVID-19.

### Sociodemographic factors

Older adults had less worry about COVID-19 in the post-acute phase, which is consistent with prior research assessing worry before infection [5,10]. In this study, newly identified contributing factors to this may be having increased trust in their regular healthcare provider and increased odds of being offered mAb treatment (per EUA guidelines [21–22]). However, patients aged 45–64 had less trust in general of providers. Further understanding is needed of the components that affect trust in these age groups.

Female patients had higher worry about COVID-19, which was true before they were infected [5,6,8,9]. This is an interesting finding given that males had worse outcomes from COVID-19 [13,42]. Female patients also had increased worry about mAbs and experienced more difficulty in receiving mAbs indicating that there may be unidentified barriers (i.e., childcare, transportation) for female patients. Addressing these barriers via policy changes, case management, and community support could enhance access to in-person treatments. Further understanding of these differences between the male and female COVID-19 patients is needed to decrease barriers to treatment.

Hispanic patients had increased worry about COVID-19 once they had been diagnosed which has been shown in some studies [11–12] prior to infection. Interestingly, Hispanic patients had higher trust in their regular provider (limited to those

**Table 4. Adjusted odds ratio for worried about, difficulty getting, recommending, and never offered mAb treatment.**

| | Received mAbs N = 539 | | | | | | Didn't Receive mAbs N = 986 | |
|---|---|---|---|---|---|---|---|---|
| | Worried Getting mAbs | | Difficultly Getting mAbs | | Recommend mAbs | | Never Offered mAbs | |
| | OR | 95% CI | OR | 95% CI | OR | 95% CI | OR | 95% CI |
| **Age in Years** | | | | | | | | |
| 18-44 years | NA | | NA | | NA | | NA | |
| 45-64 years | 0.81 | (0.42, 1.60) | 1.08 | (0.54, 2.23) | 1.13 | (0.45, 2.73) | 0.73 | (0.48, 1.09) |
| 65 + years | 0.64 | (0.31, 1.33) | 0.81 | (0.40, 1.70) | 1.52 | (0.58, 3.74) | 0.56 | (0.35, 0.89) |
| **Gender** | | | | | | | | |
| Male | NA | | NA | | NA | | NA | |
| Female | 2.3 | (1.39, 3.89) | 1.79 | (1.08, 3.00) | 0.89 | (0.46, 1.69) | 0.86 | (0.62, 1.19) |
| **Race/Ethnicity** | | | | | | | | |
| Non-Hispanic White | NA | | NA | | NA | | NA | |
| Non-Hispanic Black | 2.37 | (0.67, 7.80) | 1.55 | (0.32, 5.65) | 0.88 | (0.21, 6.15) | 0.74 | (0.39, 1.48) |
| Hispanic | 0.88 | (0.38, 1.86) | 0.73 | (0.26, 1.72) | 2.73 | (0.77, 17.39) | 1.12 | (0.70, 1.84) |
| Other | 1.13 | (0.05, 9.22) | 1.41 | (0.07, 10.45) | 0.90 | (0.10, 20.26) | 5.67 | (1.10, 104.20) |
| **Level of Education** | | | | | | | | |
| < 2 yr college | NA | | NA | | NA | | NA | |
| ≥ 2 yr college | 0.49 | (0.29, 0.83) | 0.95 | (0.54, 1.73) | 1.39 | (0.66, 2.82) | 0.82 | (0.57, 1.18) |
| **Usual Place of Care** | | | | | | | | |
| Regular Doctor | NA | | NA | | NA | | NA | |
| Walk-In Care | 0.54 | (0.24, 1.15) | 1.11 | (0.50, 2.30) | 2.76 | (0.88, 12.43) | 0.77 | (0.52, 1.16) |
| **Health Insurance** | | | | | | | | |
| Commercial/Medicare | NA | | NA | | NA | | NA | |
| Medicaid | 1.32 | (0.56, 2.94) | 0.20 | (0.03, 0.73) | 2.03 | (0.53, 13.57) | 1.88 | (1.06, 3.50) |
| None/Other | 3.03 | (0.37, 17.65) | 0.60 | (0.03, 4.13) | 0.50 | (0.07, 10.19) | 3.01 | (1.26, 8.92) |
| **Number of Comorbid Conditions** | | | | | | | | |
| None | NA | | NA | | NA | | NA | |
| One | 1.36 | (0.74, 2.50) | 1.13 | (0.63, 2.05) | 1.35 | (0.61, 2.97) | 0.70 | (0.48, 1.02) |
| Two or more | 1.24 | (0.64, 2.43) | 0.86 | (0.45, 1.67) | 1.07 | (0.48, 2.39) | 0.86 | (0.56, 1.33) |
| **Body Mass Index (BMI)** | | | | | | | | |
| BMI < 25 | NA | | NA | | NA | | NA | |
| BMI>=25 | 1.92 | (1.00, 3.93) | 0.85 | (0.48, 1.54) | 1.85 | (0.90, 3.69) | 0.77 | (0.50, 1.16) |
| **Vaccination Status** | | | | | | | | |
| No Vaccine | NA | | NA | | NA | | NA | |
| One Vaccine | 0.47 | (0.18, 1.12) | 0.42 | (0.13, 1.10) | 0.68 | (0.27, 1.94) | 1.61 | (0.92, 2.96) |
| At Least 2 Vaccines | 0.60 | (0.35, 1.04) | 0.91 | (0.53, 1.54) | 1.30 | (0.64, 2.69) | 0.61 | (0.42, 0.89) |

Legend: Responses to questions "To better understand why you did not receive the mAb treatment for COVID-19…do any of the following reasons apply to you?" with responses dichotomized into offered mAbs and not offered mAbs; "On a scale of 1 to 10, how worried were you about getting the mAb treatment?" with responses grouped into not worried (1–3), moderately worried (4–6), and highly worried (7–10); "On a scale of 1 to 10, how difficult was it for you to get the mAb treatment?" with responses grouped into not difficult (1–3), moderately difficult (4–6) and very difficult (7–10); and "If you had a close friend or relative who got COVID-19 and they were eligible to receive the mAb treatment…on a scale of 1–10, how likely would you be to recommend it? With responses grouped into not likely (1–3), moderately likely (4–6) and very likely (7–10). mAb, monoclonal antibody; OR, odds ratio; CI, confidence interval

patients who have a regular provider (79 Hispanic/243 Total)). This finding is opposite of what was seen for older patients indicating that provider trust is not the only factor impacting worry about a COVID-19 diagnosis. Previous studies demonstrate that trust is greater in race and/or language concordant patient-provider dyads [43–45] so further investigation into understanding the components of trust for racial and ethnic groups is critical. Additionally, Hispanic patients were more likely to recommend mAbs. Given that there are many barriers for minority race/ethnicity patients in accessing care [46], addressing these are critical and may have had a significant impact on receiving mAbs [47]. Further efforts to ensure equal access to COVID-19 treatment for all race/ethnicity patients are critical [48] which may include mobile units or healthcare located in community centers.

Patients with ≥ 2 years college education had increased trust in general of providers, increased trust in their regular provider, and less worry about mAbs. Patients may have been less worried about the mAbs for various reasons including their trust in their providers as well as potentially having more education about and access to the treatment. Patients who received most of their care in a walk-in clinic site were less likely to report being treated fairly, had less trust in general of providers, and less trust in their regular provider. Despite previous studies indicating patient satisfaction with walk-in clinics [49], this study indicates that emergency departments, urgent care clinics, and other walk-in clinics need to develop patient flow systems and provider education to ensure fair treatment and processes to rapidly create trust between the patient and the provider.

Patients on Medicaid insurance had increased worry about COVID-19, potentially due to financial concerns. In addition, these patients were less likely to be offered mAbs, however, they found it less difficult to receive mAbs. Medicaid's transportation support services may have contributed to these patients' ability to access treatment. Patients with none/other insurance had less trust in general of providers and were less likely to be offered mAb treatment. Insurance status has been shown to impact access to care [50] however, in this situation, Medicaid covered the full cost of mAbs and reimbursement for the treatment of uninsured patients was provided through the Health Resources & Service Administration (HRSA) COVID-19 Uninsured Program [51]. As such, it is an interesting finding that these patients were less likely to be offered treatment and indicates that there may have been gaps in provider awareness of the process to access mAb treatment and/or reimbursement, and/or bias in the treatment of Medicaid and none/other insurance patients. In the future, extensive provider educational campaigns on financial reimbursement programs could minimize this disparity.

### Clinical factors

Patients with comorbid conditions had increased worry about COVID-19 which was appropriate given that additional comorbid conditions did increase the risk for severe COVID-19 [52–53]. In addition, these patients had increased trust in general of providers which may be due to regular encounters with providers due to their comorbid conditions [54]. Patients with a BMI ≥ 25 had increased worry about COVID-19 and about mAbs. Obesity did increase the risk of severe infection [52–53] so it is congruent that these patients were more worried about COVID-19. The reasons behind their increased worry about mAbs are unclear although may have been due to the novelty of the treatment or interaction of the treatment with their medical conditions.

Patients who had received two or more COVID-19 vaccinations had increased worry about COVID-19, were more likely to be treated fairly, and have high trust in providers in general; and were more likely to be offered mAbs. Patients who received COVID-19 vaccinations had trust of providers in general which may have contributed to their willingness to get COVID-19 vaccinations. Those with two or more COVID-19 vaccinations were more likely to be treated fairly, indicating that those who were unvaccinated were not treated fairly, possibly due to provider frustration towards patients who were unvaccinated [54–56]. Those who were vaccinated were more likely to be offered mAbs which counters National Institutes of Health (NIH) prioritization guidelines when treatment supply was limited but may reflect increased trust with providers and therefore interactions with them [57]. Equitable and guideline implementation of limited resources is a challenge and

greater outreach to providers (potentially from state-based license boards) with easy access to educational resources could help alleviate some of these disparities.

The overall strength of this study is that it is a large quantitative survey on outpatients' experience with COVID-19 and exposes the differences in experience based on sociodemographic factors and clinical factors. While there has previously been focused work on Black outpatient experience with COVID-19 [12] as well as hospitalized experiences [58–60], this manuscript is novel in its expansion of explored experiences and differences between groups of patients. There are also limitations in this study. First, this study was performed in a major urban location with patients who were receiving care at two major metropolitan medical centers. As such, it may not include patient experiences in other health settings or locations throughout the country (i.e., rural). Second, the population surveyed was predominately English-speaking, with few Spanish-speaking patients. As such, it does not broadly represent the experiences of non-English speaking patients who frequently experience decreased access to care. Third, the questions included in this survey were not comprehensive to the COVID-19 experience and only targeted a few broad aspects. As such, there are gaps in understanding contributing individual and system issues. Fourth, there is a potential for a sampling/response bias given that those who chose to participate in the survey may have different opinions or experiences compared to those who declined participation in the survey. Additionally, this study excluded people who did not own or answer the phone (potentially due to health or socio-economic status) and those who are deaf. The response rate was 57% (1854/3267). Fifth, all information reported in this study was self-reported which may lead to over and underreporting biases.

In conclusion, this published report describes multiple aspects of outpatients' lived experience of COVID-19 including worry when diagnosed with COVID-19, provider-patient relationship during the diagnosis and treatment process, and treatment with neutralizing monoclonal antibodies. Future research is needed to address barriers to equitable quality medical care and health system changes are needed to reduce disparities. Our study demonstrated that there were key differences in the lived experience of COVID-19 based on sociodemographic factors and clinical factors, as well as where additional strategies are needed to improve this experience.

## Supporting information

**S1 File. Supplementary methods and tables.** Additional methods including patient selection for creation of a survey eligibility cohort, sociodemographic variable definitions, and outcome questions. S1 Table. Adjusted odds ratio for trust in doctor for COVID-19 related care and treated fairly by doctors, demographic factors only. S2 Table. Adjusted odds ratio for worried about, difficulty getting, recommending, and never offered mAb treatment, demographic factors only. S3 Table. Adjusted odds ratio for worried about COVID-19 diagnosis, demographic factors only.
(DOCX)

## Acknowledgments

**Additional Contributors:** Megan Branda, Karina Duarte, Madelyne Hull, Jennifer Peers, Julie Ressalam, Lauren Shviraga, Jeff Steele, and Heather Stocker for their assistance in project organization, survey administration, and data analysis.

## Author contributions

**Conceptualization:** Lindsey E. Fish, Samantha C. Roberts, Adit A. Ginde.

**Data curation:** Lindsey E. Fish, Tellen D. Bennett, Mika K. Hamer, Bethany M. Kwan, Seth Russell, Adane F. Wogu.

**Formal analysis:** Samantha C. Roberts, Tellen D. Bennett, Nichole E. Carlson.

**Funding acquisition:** Adit A. Ginde.

**Investigation:** Lindsey E. Fish, Samantha C. Roberts, Adit A. Ginde.

**Methodology:** Adit A. Ginde.

**Supervision:** Nichole E. Carlson, Adit A. Ginde.

**Visualization:** Mika K. Hamer, Bethany M. Kwan, Matthew K. Wynia, Adit A. Ginde.

**Writing – original draft:** Lindsey E. Fish, Samantha C. Roberts.

**Writing – review & editing:** Tellen D. Bennett, Nichole E. Carlson, Mika K. Hamer, Bethany M. Kwan, Seth Russell, Adane F. Wogu, Matthew K. Wynia, Adit A. Ginde.

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
