## [Decision Letter · Decision Letter 0]

21 Jan 2025

PONE-D-24-35333Experiences of Recently Diagnosed COVID-19 Outpatients: A Survey on Patient Worries, Provider-Patient Interactions and Neutralizing Monoclonal Antibody TreatmentPLOS ONE

Dear Dr. Fish,

Thank you for submitting your manuscript to PLOS ONE. After careful consideration, we feel that it has merit but does not fully meet PLOS ONE’s publication criteria as it currently stands. Therefore, we invite you to submit a revised version of the manuscript that addresses the points raised during the review process.

We look forward to receiving your revised manuscript.

Kind regards,

Ranjan K. Mohapatra, PhD

Academic Editor

PLOS ONE

Journal Requirements:

2. In the ethics statement in the Methods, you have specified that verbal consent was obtained. Please provide additional details regarding how this consent was documented and witnessed, and state whether this was approved by the IRB

“Funding for this project was by the National Center for Advancing Translational Sciences of the National Institutes of Health [grant numbers UL1TR002525, UL1TR002535-03S3, UL1TR002535-04S2, and 1UM1TR004399-01A1] and supported by the Health Data Compass Data Warehouse project (healthdatacompass.org).”

“I have read the journal's policy and the authors of this manuscript have the following competing interests: Dr. Ginde reports grants from the National Institutes of Health (NIH) for the conduct of this study and other grants related to COVID-19 from the NIH, Centers for Disease Control and Prevention, Department of Defense, AbbVie, and Faron Pharmaceuticals, outside the submitted work. The other authors have no competing interests to report.”

Please respond by return email with your amended Competing Interests Statement and we will change the online submission form on your behalf.

5. We note that you have indicated that there are restrictions to data sharing for this study. For studies involving human research participant data or other sensitive data, we encourage authors to share de-identified or anonymized data. However, when data cannot be publicly shared for ethical reasons, we allow authors to make their data sets available upon request. For information on unacceptable data access restrictions, please see http://journals.plos.org/plosone/s/data-availability#loc-unacceptable-data-access-restrictions.

6. Please include captions for your Supporting Information files at the end of your manuscript, and update any in-text citations to match accordingly. Please see our Supporting Information guidelines for more information: http://journals.plos.org/plosone/s/supporting-information .

Reviewers' comments:

Reviewer's Responses to Questions

**Comments to the Author**

1. Is the manuscript technically sound, and do the data support the conclusions?

Reviewer #1: Partly

Reviewer #2: Yes

2. Has the statistical analysis been performed appropriately and rigorously? 

Reviewer #1: No

Reviewer #2: Yes

3. Have the authors made all data underlying the findings in their manuscript fully available?

Reviewer #1: Yes

Reviewer #2: Yes

4. Is the manuscript presented in an intelligible fashion and written in standard English?

Reviewer #1: Yes

Reviewer #2: Yes

5. Review Comments to the Author

Reviewer #1: I would like to thank you for the opportunity to review this paper.

First

Please insert in the text a line numbering (continuous) with a line spacing of 12 except for ( tables and figures) when you are submitting your revised manuscript for ease of tracking of change.

Title

1. Include the study setting in the title.

Abstract

I am wondering if this is the abstract format for PLOS ONE. The abstract lacks an in-depth summary of the study.

1. Cross-check the acronym for the word “neutralizing monoclonal antibodies” I am not sure if it is (mAb) but rather (nMAbs).

2. The description of methods lacks sufficient depth.

3. The study design is a "longitudinal prospective cohort," but you failed to tell us how longitudinal data was collected to measure worry, altered provider-patient interactions, and options to use novel treatments.

4. The key results are dense and not well contextualized. Rewrite these results "Patients of younger age, female sex, Hispanic ethnicity... to “Greater worry was reported among younger patients, females, Hispanic individuals, Medicaid beneficiaries, and those with ≥2 comorbidities and BMI >25.”

5. Rewrite the conclusion to reflect your results.

6. Please I suggest you rewrite the abstract.

Introduction

1. Provide a reference (s) to these sentences “Patients with recently diagnosed COVID-19 have varied experiences of emotional worry, interactions with health care professionals, and considerations regarding novel treatments such as neutralizing monoclonal antibodies (mAbs).”

Method and materials

1. Rewrite the outcomes section of the method to improve the manuscript and ensure clarity.

2. The BMI classification. What criteria does it based on?

Results

1. For the phrase “more likely to be non-Hispanic White (81.6% vs. 71.2%)”, I suggest including the total crosstab presentation of mAb treated and non-mAb treated in Table.

2. Tables 2 and 3. Part of the tables were cut off so I cannot access the full tables.

Reviewer #2: The manuscript (PONE-D-24-35333) offers a thorough summary of the study, encompassing its objectives, methodology, key findings, and conclusions. It effectively highlights the study's significance by connecting patient experiences, healthcare disparities, and treatment accessibility within the context of COVID-19. The study design is meticulously documented, providing clear details on participant selection, survey implementation, and statistical analyses. The manuscript may be accepted after major revision.

Revisions Needed:

1. Streamline content to emphasize critical findings (e.g., key socio-demographic factors).

2. Avoid redundant phrases like “Limited research has been performed...” which i

3. s already implicit in the introduction.

4. Include actionable insights to make the conclusion more impactful.

5. Condense background information to avoid over-explaining concepts such as monoclonal antibody (mAb) efficacy.

6. Clarify the novelty of the study by explicitly contrasting it with prior research.

7. Focus on the specific research gap that this study addresses.

8. Provide more detail on survey development, including pilot testing outcomes and psychometric validation.

9. Clarify the rationale for categorization and thresholds (e.g., Likert scale recategorization from 1-10 to 1-3, 4-6, and 7-10).

10. Discuss potential biases introduced by reliance on telephone surveys and verbal consent.

11. Improve clarity by summarizing findings for each outcome variable in dedicated subheadings (e.g., “Trust in Providers,” “Worry About COVID-19”).

12. Address potential confounding variables, particularly for outcomes related to mAb treatment and worry levels.

13. Enhance the discussion on why specific sociodemographic groups (e.g., vaccinated patients) reported higher worry levels.

14. Provide more actionable solutions to address disparities (e.g., interventions to improve trust in walk-in clinics or reduce barriers for Medicaid patients).

15. Address broader implications of findings, such as policy-level recommendations for equitable healthcare delivery.

16. Ensure all figures and tables are self-explanatory by adding detailed captions.

17. Simplify complex visualizations, particularly in Figures 1 and 2, to enhance readability.

18. Add comparative summaries in tables for quicker reference.

19. Edit for grammatical consistency and conciseness. For example, “Patients of younger age, female sex…” could be simplified to “Younger and female patients.”

20. Avoid repetitive phrases and overuse of technical jargon that could be streamlined for readability.

21. Use active voice where appropriate to improve flow (e.g., “We conducted a survey” instead of “A survey was conducted”).

22. Expand on how the urban-centric and English-speaking population limits the applicability of findings to diverse settings.

23. Address the limitation of self-reported data and potential underreporting or over-reporting biases.

24. Ensure all references are correctly formatted according to journal guidelines.

6. PLOS authors have the option to publish the peer review history of their article (what does this mean? ). If published, this will include your full peer review and any attached files.

**Do you want your identity to be public for this peer review?** For information about this choice, including consent withdrawal, please see our Privacy Policy .

Reviewer #1: **Yes: ** Lawrence Sena Tuglo

Reviewer #2: No

---

## [Author Response · Author response to Decision Letter 1]

23 Mar 2025

Lindsey E. Fish MD

Samantha C. Roberts MS, MPH

March 7, 2025

Ranjan K. Mohapatra, PhD

Academic Editor

PLOS ONE

Dear Dr. Mohapatra,

Thank you so much for taking the time to review our manuscript entitled “Experiences of Recently Diagnosed COVID-19 Outpatients: A Survey on Patient Worries, Provider-Patient Interactions and Neutralizing Monoclonal Antibody Treatment” (PONE-D-24-35333). We greatly appreciate the opportunity to submit a revised version of the manuscript for further review. Please see below for our responses to each point raised by the academic editor and reviewers.

As clarification on the funding and our financial disclosure, please revise to state “Funded by the National Center for Advancing Translational Sciences of the National Institutes of Health [grant numbers UL1TR002525, UL1TR002535-03S3, UL1TR002535-04S2, and 1UM1TR004399-01A1] and supported by the Health Data Compass Data Warehouse project (healthdatacompass.org). The funders had no role in study design, data collection and analysis, decision to publish, or preparation of the manuscript.”

As clarification on the competing interest statement, please revise to state “I have read the journal's policy and the authors of this manuscript have the following competing interests: Dr. Ginde reports grants from the National Institutes of Health (NIH) for the conduct of this study and other grants related to COVID-19 from the NIH, Centers for Disease Control and Prevention, Department of Defense, AbbVie, and Faron Pharmaceuticals, outside the submitted work. The other authors have no competing interests to report. This does not alter our adherence to PLOS ONE policies on sharing data and materials.”

From reviewers:

-Thank you for this suggestion. We have ensured that the manuscript meets PLOS ONE’s style requirements and file naming.

2. In the ethics statement in the Methods, you have specified that verbal consent was obtained. Please provide additional details regarding how this consent was documented and witnessed, and state whether this was approved by the IRB

- Thank you for this suggestion. We have provided additional details in this section.

“Funding for this project was by the National Center for Advancing Translational Sciences of the National Institutes of Health [grant numbers UL1TR002525, UL1TR002535-03S3, UL1TR002535-04S2, and 1UM1TR004399-01A1] and supported by the Health Data Compass Data Warehouse project (healthdatacompass.org).”

- Thank you for this suggestion. We have added the necessary additional documentation, "The funders had no role in study design, data collection and analysis, decision to publish, or preparation of the manuscript."

“I have read the journal's policy and the authors of this manuscript have the following competing interests: Dr. Ginde reports grants from the National Institutes of Health (NIH) for the conduct of this study and other grants related to COVID-19 from the NIH, Centers for Disease Control and Prevention, Department of Defense, AbbVie, and Faron Pharmaceuticals, outside the submitted work. The other authors have no competing interests to report.”

Please respond by return email with your amended Competing Interests Statement and we will change the online submission form on your behalf.

- Thank you for this suggestion. We have added the necessary additional documentation, “This does not alter our adherence to PLOS ONE policies on sharing data and materials.”

5. We note that you have indicated that there are restrictions to data sharing for this study. For studies involving human research participant data or other sensitive data, we encourage authors to share de-identified or anonymized data. However, when data cannot be publicly shared for ethical reasons, we allow authors to make their data sets available upon request. For information on unacceptable data access restrictions, please see http://journals.plos.org/plosone/s/data-availability#loc-unacceptable-data-access-restrictions.

- Thank you for the request for clarification. We can collaborate with someone external via a process that entails a reasonable request and determination by the study steering committee. However, we cannot share data due to data sharing policies and data use agreements in place with participating sites. Data requests should be sent to Dr. Adit A. Ginde, adit.ginde@cuanschutz.edu.

- There are restrictions and as such, we are unable to upload the data as stated above.

- Thank you for this suggestion. There was no change to the Data Availability statement in the submission, however we did add more information for data sharing requests.

- Thank you for this suggestion. We have updated our files.

Reviewers' comments:

5. Review Comments to the Author

Reviewer #1: I would like to thank you for the opportunity to review this paper.

First

Please insert in the text a line numbering (continuous) with a line spacing of 12 except for ( tables and figures) when you are submitting your revised manuscript for ease of tracking of change.

- Thank you for this suggestion. We have made this change.

Title

1. Include the study setting in the title.

- Thank you for this suggestion. We have added the setting.

Abstract

I am wondering if this is the abstract format for PLOS ONE. The abstract lacks an in-depth summary of the study.

- Thank you for this suggestion. We have revised the abstract.

1. Cross-check the acronym for the word “neutralizing monoclonal antibodies” I am not sure if it is (mAb) but rather (nMAbs).

- Thank you for this suggestion. Both acronyms have been used, however, we have chosen to continue using mAb.

2. The description of methods lacks sufficient depth.

- Thank you for this suggestion. We have updated the primary manuscript methods to include more depth which was previously in the supplemental materials.

3. The study design is a "longitudinal prospective cohort," but you failed to tell us how longitudinal data was collected to measure worry, altered provider-patient interactions, and options to use novel treatments.

- Thank you so much for this request for clarification. We have clarified this in the manuscript.

4. The key results are dense and not well contextualized. Rewrite these results "Patients of younger age, female sex, Hispanic ethnicity... to “Greater worry was reported among younger patients, females, Hispanic individuals, Medicaid beneficiaries, and those with ≥2 comorbidities and BMI >25.”

- Thank you for this suggestion. We have revised the results.

5. Rewrite the conclusion to reflect your results.

- Thank you for this suggestion. We have revised the conclusion.

6. Please I suggest you rewrite the abstract.

- Thank you for this suggestion. We have revised the abstract.

Introduction

1. Provide a reference (s) to these sentences “Patients with recently diagnosed COVID-19 have varied experiences of emotional worry, interactions with health care professionals, and considerations regarding novel treatments such as neutralizing monoclonal antibodies (mAbs).”

- Thank you for this suggestion. We have added the references.

Method and materials

1. Rewrite the outcomes section of the method to improve the manuscript and ensure clarity.

- Thank you for this suggestion. We have revised the outcomes section.

2. The BMI classification. What criteria does it based on?

- This was based on standard calculations as performed in the Electronic Health Record. We have added clarification to the manuscript.

Results

1. For the phrase “more likely to be non-Hispanic White (81.6% vs. 71.2%)”, I suggest including the total crosstab presentation of mAb treated and non-mAb treated in Table.

- Thank you very much for this suggestion. We have included this in Table 1.

2. Tables 2 and 3. Part of the tables were cut off so I cannot access the full tables.

- Thank you for letting us know. We are not sure what happened but will ensure everything is visible in the resubmission.

Reviewer #2: The manuscript (PONE-D-24-35333) offers a thorough summary of the study, encompassing its objectives, methodology, key findings, and conclusions. It effectively highlights the study's significance by connecting patient experiences, healthcare disparities, and treatment accessibility within the context of COVID-19. The study design is meticulously documented, providing clear details on participant selection, survey implementation, and statistical analyses. The manuscript may be accepted after major revision.

Revisions Needed:

1. Streamline content to emphasize critical findings (e.g., key socio-demographic factors).

- Thank you for this suggestion. We have revised the discussion to highlight critical findings.

2. Avoid redundant phrases like “Limited research has been performed...” which i

s already implicit in the introduction

- Thank you for this suggestion. We have revised the introduction.

3. Include actionable insights to make the conclusion more impactful.

- Thank you for this suggestion. We have revised the discussion.

4. Condense background information to avoid over-explaining concepts such as monoclonal antibody (mAb) efficacy.

- Thank you for this suggestion. We have revised the introduction.

5. Clarify the novelty of the study by explicitly contrasting it with prior research.

- Thank you for this suggestion. We have revised the discussion.

6. Focus on the specific research gap that this study addresses.

- Thank you for this suggestion. We have provided clarification.

7. Provide more detail on survey development, including pilot testing outcomes and psychometric validation.

- Thank you for this suggestion. We have provided additional information.

8. Clarify the rationale for categorization and thresholds (e.g., Likert scale recategorization from 1-10 to 1-3, 4-6, and 7-10).

- Thank you for this suggestion. We have provided additional information.

9. Discuss potential biases introduced by reliance on telephone surveys and verbal consent.

- Thank you for this suggestion. We have provided additional information.

10. Improve clarity by summarizing findings for each outcome variable in dedicated subheadings (e.g., “Trust in Providers,” “Worry About COVID-19”).

- Thank you for this suggestion. We have added subheadings to the discussion.

11. Address potential confounding variables, particularly for outcomes related to mAb treatment and worry levels.

- Thank you for this suggestion. We have addressed potential confounding.

12. Enhance the discussion on why specific sociodemographic groups (e.g., vaccinated patients) reported higher worry levels.

-Thank you for this suggestion. We have added this to the discussion.

13. Provide more actionable solutions to address disparities (e.g., interventions to improve trust in walk-in clinics or reduce barriers for Medicaid patients).

-Thank you for this suggestion. We have added this to the discussion.

14. Address broader implications of findings, such as policy-level recommendations for equitable healthcare delivery.

- Thank you for this suggestion. We have added this to the discussion.

15. Ensure all figures and tables are self-explanatory by adding detailed captions.

- Thank you for this suggestion. We have revised the captions for the tables and figures.

16. Simplify complex visualizations, particularly in Figures 1 and 2, to enhance readability.

- Thank you for this suggestion. We improved the figures.

17. Add comparative summaries in tables for quicker reference.

- Thank you for this suggestion. We have chosen to keep the comparative summary in the results section to keep the tables simpler.

18. Edit for grammatical consistency and conciseness. For example, “Patients of younger age, female sex…” could be simplified to “Younger and female patients.”

- Thank you for this suggestion. We have incorporated this throughout the manuscript.

19. Avoid repetitive phrases and overuse of technical jargon that could be streamlined for readability.

- Thank you for this suggestion. We have incorporated this throughout the manuscript.

20. Use active voice where appropriate to improve flow (e.g., “We conducted a survey” instead of “A survey was conducted”).

- Thank you for this suggestion. We have incorporated this throughout the manuscript.

21. Expand on how the urban-centric and English-speaking population limits the applicability of findings to diverse settings.

- Thank you for this suggestion. We have added this to the discussion.

22. Address the limitation of self-reported data and potential underreporting or over-reporting biases.

- Thank you for this suggestion. We have added this information.

23. Ensure all references are correctly formatted according to journal guidelines.

- Thank you for this suggestion. We have correctly formatted the references.

Thank you for much for the opportunity to submit a revision.

Sincerely,

Lindsey E Fish MD

Samantha C. Roberts MS, MPH

---

## [Decision Letter · Decision Letter 1]

23 May 2025

Experiences of recently diagnosed urban COVID-19 outpatients: a survey on patient worries, provider-patient interactions and neutralizing monoclonal antibody treatment

PONE-D-24-35333R1

Dear Dr. Fish,

We’re pleased to inform you that your manuscript has been judged scientifically suitable for publication and will be formally accepted for publication once it meets all outstanding technical requirements.

Kind regards,

Ranjan K. Mohapatra, PhD

Academic Editor

PLOS ONE

Additional Editor Comments (optional):

Reviewers' comments:

Reviewer's Responses to Questions

**Comments to the Author**

1. If the authors have adequately addressed your comments raised in a previous round of review and you feel that this manuscript is now acceptable for publication, you may indicate that here to bypass the “Comments to the Author” section, enter your conflict of interest statement in the “Confidential to Editor” section, and submit your "Accept" recommendation.

Reviewer #1: All comments have been addressed

Reviewer #2: All comments have been addressed

2. Is the manuscript technically sound, and do the data support the conclusions?

Reviewer #1: Yes

Reviewer #2: Yes

3. Has the statistical analysis been performed appropriately and rigorously? 

Reviewer #1: Yes

Reviewer #2: I Don't Know

4. Have the authors made all data underlying the findings in their manuscript fully available?

Reviewer #1: Yes

Reviewer #2: Yes

5. Is the manuscript presented in an intelligible fashion and written in standard English?

Reviewer #1: Yes

Reviewer #2: Yes

6. Review Comments to the Author

Reviewer #1: I recommend acceptance because the authors have addressed my comments.I recommend acceptance because the authors have addressed my comments. I recommend acceptance.

Reviewer #2: The study provides significant insights into the experiences of COVID-19 outpatients and contributes valuable information on an important public health topic. Given the satisfactory revisions and the importance of the subject matter, I recommend that the manuscript be accepted for publication in PLOS ONE. Additionally look for the references style and other things to rule out any errors that are not in the preview of journal’s guidelines.

7. PLOS authors have the option to publish the peer review history of their article (what does this mean? ). If published, this will include your full peer review and any attached files.

**Do you want your identity to be public for this peer review?** For information about this choice, including consent withdrawal, please see our Privacy Policy .

Reviewer #1: **Yes: ** Lawrence Sena Tuglo

Reviewer #2: **Yes: ** Dr. Puneet Kumar Singh

---

## [Editor Report · Acceptance letter]

PONE-D-24-35333R1

PLOS ONE

Dear Dr. Fish,

I'm pleased to inform you that your manuscript has been deemed suitable for publication in PLOS ONE. Congratulations! Your manuscript is now being handed over to our production team.

Kind regards,

on behalf of

Dr. Ranjan K. Mohapatra

Academic Editor

PLOS ONE